# Exposure to duloxetine during pregnancy and risk of congenital malformations and stillbirth: A nationwide cohort study in Denmark and Sweden

Mikkel Zöllner Ankarfeldt[1]*, Janne Petersen[1,2], Jon Trærup Andersen[3,4], Hu Li[5], Stephen Paul Motsko[5], Thomas Fast[6], Simone Møller Hede[6], Espen Jimenez-Solem[1,3,4]

1 Copenhagen Phase IV Unit (Phase4CPH), Department of Clinical Pharmacology and Center for Clinical Research and Prevention, Copenhagen University Hospital Bispebjerg and Frederiksberg, Copenhagen, Denmark, 2 Section of Biostatistics, Department of Public Health, University of Copenhagen, Copenhagen, Denmark, 3 Department of Clinical Pharmacology, Copenhagen University Hospital Bispebjerg and Frederiksberg, Copenhagen, Denmark, 4 Faculty of Health and Medical Sciences, University of Copenhagen, Copenhagen, Denmark, 5 Eli Lilly and Company, Indianapolis, Indiana, United States of America, 6 Institute of Applied Economics & Health Research, Copenhagen, Denmark

* mikkelza@gmail.com

**Data Availability Statement:** Data from the Danish and Swedish registers are third party data, meaning that we as researchers do not hold the

## Abstract

### Background

The prevalence of depression and the exposure to antidepressants are high among women of reproductive age and during pregnancy. Duloxetine is a selective serotonin-norepinephrine reuptake inhibitor (SNRI) approved in the United States and Europe in 2004 for the treatment of depression. Fetal safety of duloxetine is not well established. The present study evaluates the association of exposure to duloxetine during pregnancy and the risk of major and minor congenital malformations and the risk of stillbirths.

### Methods and findings

A population-based observational study was conducted based on data from registers in Sweden and Denmark. All registered births and stillbirths in the medical birth registers between 2004 and 2016 were included. Malformation diagnoses were identified up to 1 year after birth. Logistic regression analyses were used. Potential confounding was addressed through multiple regression, propensity score (PS) matching, and sensitivity analyses. Confounder variables included sociodemographic information (income, education, age, year of birth, and country), comorbidity and comedication, previous psychiatric contacts, and birth-related information (smoking during pregnancy and previous spontaneous abortions and stillbirths).

Duloxetine-exposed women were compared with 4 comparators: (1) duloxetine-nonexposed women; (2) selective serotonin reuptake inhibitor (SSRI)-exposed women; (3) venlafaxine-exposed women; and (4) women exposed to duloxetine prior to, but not during, pregnancy. Exposure was defined as redemption of a prescription during the first trimester

data, but have obtained data after application at relevant parties. The Danish data can be applied for at Statistics Denmark (https://www.dst.dk/en/TilSalg/Forskningsservice). The Swedish data can be applied for at Statistic Sweden (https://www.scb.se/en/About-us/contact-us/) and Swedish National Board of Health and Welfare data (https://www.socialstyrelsen.se/en/statistics-and-data/statistics/statistical-databases/).

**Funding:** The study was performed by the Copenhagen Phase 4 Unit (Phase4CPH), Department of Clinical Pharmacology and the Institute of Applied Economics and Health Research Aps (ApHER), and financed by Eli Lilly, the manufacturer of duloxetine. MZA and JP are employed at the Copenhagen Phase 4 Unit (Phase4CPH). JTA and EJS are employed at the Department of Clinical Pharmacology. HL and SPM are former employees of Eli Lilly and Company and thereby received a salary from the funder. HL is currently employed by Gilead Science Inc. and SPM is currently employed by Amgen Inc. TF is former employee of ApHER and is currently employed by Quantify Research. SMH is employed at ApHER. The funder had the opportunity to comment study design, data collection and analysis, where to publish and preparation of the manuscript, but MZA and EJS had the final decisions.

**Competing interests:** I have read the journal's policy and the authors of this manuscript have the following competing interests: MZA, JP, and EJS have performed other studies regarding antidepressants involving funding from Janssen Pharmaceutical via Phase4CPH. JP is also supervising a PhD student in the area of pregnancy outcomes and insulin funded by Novo Nordisk A/S. JTA, TF, and SMH have no relevant financial activities outside the submitted work in the past 36 months. HL and SPM are former employees of Eli Lilly and Company and are minor stockholders.

**Abbreviations:** CI, confidence interval; ENCePP, European Network of Centres for Pharmacoepidemiology and Pharmacovigilance; LMP, last menstrual period; OR, odds ratio; PS, propensity score; SGA, small for gestational age.

and throughout pregnancy for the analyses of malformations and stillbirths, respectively. Outcomes were major and minor malformations and stillbirths gathered from the national patient registers. The cohorts consisted of more than 2 million births with 1,512 duloxetine-exposed pregnancies. No increased risk for major malformations, minor malformations, or stillbirth was found across comparison groups in adjusted and PS-matched analyses. Duloxetine-exposed versus duloxetine-nonexposed PS-matched analyses showed odds ratio (OR) 0.98 (95% confidence interval [CI] 0.74 to 1.30, $p = 0.909$) for major malformations, OR 1.09 (95% CI 0.82 to 1.45, $p = 0.570$) for minor malformation, and 1.18 (95% CI 0.43 to 3.19, $p = 0.749$) for stillbirths. For the individual malformation subtypes, some findings were statistically significant but were associated with large statistical uncertainty due to the extremely small number of events. The main limitations for the study were that the indication for duloxetine and a direct measurement of depression severity were not available to include as covariates.

## Conclusions

Based on this observational register-based nationwide study with data from Sweden and Denmark, no increased risk of major or minor congenital malformations or stillbirth was associated with exposure to duloxetine during pregnancy.

## Author summary

### Why was this study done?

- Many women of reproductive age take drugs used to treat depression, including duloxetine, a selective serotonin-norepinephrine reuptake inhibitor (SNRI) approved in 2004 for the treatment of depression, and the use of which has been increasing.

- There is a need for information on the possible association between exposure to duloxetine and malformation or stillbirth among offspring.

### What did the researchers do and find?

- In a nationwide register-based study, all women with pregnancies ending in a live birth or stillbirth in Denmark and Sweden were analyzed.

- From more than 2 million births identified, information on drug exposure, comorbidities, education and income, and congenital malformations and stillbirths was gathered.

- The analyses taking into account factors beyond duloxetine exposure did not reveal associations between exposure to duloxetine during pregnancy and risk for malformations or stillbirth.

**What do these findings mean?**

- This study with nationwide register data from Sweden and Denmark found no increased risk of congenital major or minor malformations or stillbirths among women exposed to duloxetine during pregnancy.

- The study analyzes the risk of malformations and stillbirths. Other safety outcomes (e.g., preterm birth or small for gestational age) were not addressed and need to be analyzed in future studies.

## Introduction

Depression or depressive symptoms are common during pregnancy [1–4]. Despite a drop in recent years [5], the use of antidepressants among pregnant women has grown steadily [6–12]. Selective serotonin reuptake inhibitors (SSRIs) are the most common [9,12,13], followed by serotonin and norepinephrine reuptake inhibitors (SNRIs) [5,12,14].

Duloxetine (an SNRI) was approved in the United States and Europe in 2004. In Europe, the indication is for major depressive disorder, generalized anxiety disorder, stress urinary incontinence, and diabetic peripheral neuropathy. A common indication for women of child-bearing age is for depressive disorder [15]. The safety of antidepressants during pregnancy, especially their teratogenic effect, has been uncertain [16–20]. However, studies that address potential confounding report no association between first trimester exposure to SSRIs and malformation [21] or stillbirth [22,23]. This is not necessarily applicable to SNRIs, since they affect also norepinephrine levels [24]. Studies on SNRIs suggest no increased risk of major malformation, based on postmarketing surveillance systems [25–27], small cohorts without a comparison group [28,29], or cohorts with a comparison group [30–32]. A recent review found no increased risk of major malformations but concluded that the evidence for duloxetine is limited [33]. A large cohort study is necessary to assess the risk of rare outcomes (e.g., malformations, stillbirth).

The present study evaluates the association between duloxetine exposure during pregnancy and the risk of major and minor congenital malformation and stillbirths in a cohort based on all pregnancies in Sweden and Denmark between 2004 and 2016.

## Methods

The present study is based on a safety study regarding duloxetine and pregnancy outcomes, with the protocol and the full study report available via the European Network of Centres for Pharmacoepidemiology and Pharmacovigilance (ENCePP, EUPAS20253) [34]. Beside malformation and stillbirth, the protocol and full study include abortion, preterm birth, and being born small for gestational age (SGA) as outcomes. Results about abortion is published elsewhere [35], and results about preterm birth and SGA is under preparation for publication.

The study is based on nationwide registers from Sweden and Denmark covering all registered births from 2004 to 2016, with 1-year follow-up data of congenital malformations. Registers were linked with unique personal identification numbers given to all Swedish and Danish citizens upon birth or immigration. The following Danish and Swedish nationwide registers were used. The prescription registers [36–38], containing electronically submitted information on prescriptions dispensed by pharmacies, classified according to the global ATC system. The patient registers [39,40] that include discharge diagnoses of all inpatients and outpatients in

contact with a hospital. The medical birth registers [41–43] were all live births as well as still-births from varying gestational ages in the different countries are notified to the registers with information on the mother, the neonate, and the father as well. Registers holding information about education and household income [44–46] based on national statistics on education (highest obtained education) and annual tax reports.

The study was approved by the Swedish regional ethics review board in Gothenburg (ref: 1040–17 and T782-18), the Swedish National Board of Health and Welfare (ref: 30714/2017), and in Denmark by the Data Protection Agency (j.nr. VD-2018-371, I-Suite nr. 6621). No approval from the Danish Research Ethics Committees for the Capital Region was needed since only national registers were used.

## Cohorts

The cohort consisted of registered live and stillbirths from 2004 to 2016 of women with a valid personal identification number aged 18 and above. Exclusion criteria for the malformation analyses were the following: mothers migrating between 365 days prior to last menstrual period (LMP) until 365 days postdelivery, stillbirths, invalid personal identification number of offspring, births with a chromosomal abnormality diagnosis (ICD-10 codes Q87.1, Q87.4, Q9X), and mothers with a redeemed prescription for a teratogenic drug in the period from LMP to 90 days post LMP (warfarin [ATC: B01AA03], antineoplastic agents [ATC: L01], iso-tretinoin [ATC: D10AD04, D10BA01, D10AD54], misoprostol [ATC: A02BB01, G02AD06, M01AE56], lithium [ATC: N05AN01], and thalidomide [ATC: L04AX02]). Exclusion criteria for the stillbirth analyses were the following: mothers migrated between 90 days prior to LMP until delivery and gestational age shorter than 22 weeks or longer than 45 weeks.

## Exposure, comparison groups, and outcome

Maternal exposure to medication was defined as a redeemed prescription at a pharmacy. The exposure time window for malformations was from LMP to 90 days after LMP, corresponding to the first trimester of the pregnancy. The exposure time window for stillbirth was from LMP to end of pregnancy. With maternal exposure, fetal exposure is assumed.

Duloxetine exposure was defined as at least one redeemed prescription of duloxetine (ATC N06AX21) in the exposure time window. Four comparison groups were used, all with no redeemed prescription of duloxetine in the relevant exposure time window: (1) duloxetine nonexposed: no redeemed prescription of duloxetine in the exposure time window; (2) SSRI exposed: at least one redeemed prescription of an SSRI (ATC N06AB) in the exposure time window; (3) venlafaxine exposed: at least one redeemed prescription of venlafaxine (ATC N06AX16. Venlafaxine is an SNRI like duloxetine) in the exposure time window; and (4) duloxetine discontinuers: at least one redeemed prescription of duloxetine between 365 days prior to LMP to LMP and not during pregnancy. SSRI-exposed and venlafaxine-exposed women and duloxetine discontinuers were used as comparators to take confounding by indi-cation and severity, and maybe even unmeasured confounding, into account, as they are expected to be similar to duloxetine-exposed women with regard to, e.g., the underlying psy-chological disease, comorbidity, and health behavior. The comparison groups were not mutu-ally exclusive, and comparisons were analyzed separately with each comparison group. For both the malformation and the stillbirth analyses, an additional exclusion criterion was applied when comparing duloxetine-exposed women with duloxetine nonexposed, SSRI exposed, and venlafaxine exposed: women with duloxetine exposure from 90 days prior to LMP but no exposure from LMP to 90 days after LMP were excluded. This washout period was applied to avoid misclassification.

Major and minor malformations were classified according to the EUROCAT classification of congenital malformations version 1.4 [47]. Diagnoses of the offspring were gathered from the national patient registers as either a primary or secondary diagnosis registered within 365 days after birth. Major malformations were defined as the following ICD-10 codes: Q-chapter, D215, D821, D1810, P350, P351, P371, except for the ICD-10 codes used to define minor malformations. Minor malformations were defined as the ICD-10 codes in Table A in S1 Tables. Also, analyses of major malformation subtypes were performed: abdominal wall; cardiac; digestive system; ear, face, and neck; eye; genital; limb; nervous system; orofacial clefts; respiratory system; urinary system; and other anomalies (Table B in S1 Tables). Information on stillbirth was gathered from the medical birth registers and was defined as no signs of life at birth after week 22 of pregnancy and from the patient register if abortions were registered after week 22 [48].

## Statistical methods

For each of the 4 comparator groups, an unadjusted, an adjusted multiple regression, and a propensity score (PS)-matched analysis were performed.

The PSs were estimated with logistic regression and greedy matching using the SAS macro *OneToManyMTCH* [49] with an extension to secure only women with a difference of maximum of 0.2 logit of the PS were used to match. When matching duloxetine exposed with duloxetine nonexposed, a 1:4 ratio was used. Because of limited data in the other comparison groups, duloxetine and SSRI exposed were matched with a 1:2 ratio and venlafaxine exposed and duloxetine discontinuers with a 1:1 ratio. Duloxetine-exposed individuals with no match were excluded from the PS-matched analyses. After PS matching, a conditional logistic regression, including the matched group id as a strata variable, was fitted to assess risk of minor and major malformations and stillbirths, respectively. To assess the balance of possible confounders after PS matching, standardized differences were calculated using the SAS macro *stddiff macro* [50].

For the adjusted multiple regression and the PS, prespecified covariates were used based on previous publications [21,23] and available data. When fitting each model, covariates were removed, if the model could not be estimated. This is the case if, e.g., no patients in one exposure group had any severe stress reaction. Then, the variable severe stress reaction cannot be part of the model and the covariate needs to be removed.

Covariates used for the malformation and stillbirth analyses are the following: data source (Sweden/Denmark), birth year of the offspring (3 categories: 2004 to 2008, 2009 to 2012, and 2013 to 2016), maternal age (4 categories: 18 to 24, 25 to 29, 30 to 34, and >34 years), previous spontaneous abortions (0/1/≥2), previous stillbirths (yes/no), smoking during pregnancy (yes/no), psychiatric hospitalizations (1 year prior to LMP: yes/no), psychiatric outpatient visits (1 year prior to LMP: yes/no), household income (year of LMP, grouped in quartiles), and highest completed education (year of LMP, 3 categories: <11, 11 to 15, and >15 years). Comorbidities (identified up to maximum 5 years prior to LMP; see Table C in S1 Tables for ICD-10 codes and ATC codes): affective disorder, anxiety or phobia, depression, diabetes during pregnancy, diabetes, diabetic peripheral neuropathy, hyperthyroidism and hypothyroidism, hypertension, obesity, renal failure, severe stress reaction, and stress urinary incontinence. Comedication (at least one redeemed prescription between 90 days prior to LMP to end of the relevant exposure time window; see Table D in S1 Tables for ATC codes): antiepileptics, antihypertensives, antipsychotics, anxiolytics, danazol, estradiol, fluconazole, glucose-lowering, NSAID, opioids, progesterone, steroid hormone, thyroid hormone, and triptans. SSRI and venlafaxine comedication were not part of the confounder selection, since these will be highly associated with the

comparison group rather than the duloxetine exposed. For a complete list of covariates used for the individual analysis, see Table E in S1 Tables.

In general, there were very few missing values. If income was missing at year of LMP, first income was imputed from 1 year prior to LMP, and, if still missing, income was imputed from 1 year after LMP, where possible. If education was missing at year of LMP, it was imputed from 1 year after LMP, where possible. Data were analyzed assuming missing at random and persons with missing values were deleted from the analysis.

For the analyses of stillbirth and malformation subtypes with less than 30 outcome events in the exposed group, only unadjusted and PS-matched analyses were performed.

Four prespecified sensitivity analyses were conducted. To assess the exposure definition of minimum one redeemed prescription and evaluate potential misclassification of exposure, sensitivity analyses were conducted with (1) exposure redefined to minimum 2 redeemed prescriptions; and (2) exposure redefinition to overlap between the exposure time window and days' supply of redeemed prescriptions. Days' supply was based on the number and strength of redeemed pills compared with the WHO's daily defined dose [51]. A woman's first pregnancy may influence later pregnancies; therefore, sensitivity analyses (3) restricting the cohort to the first pregnancy within the study period were performed. The medical birth registers hold information on maternal BMI, but this is expected to be missing for a substantial number of women. The main analyses did not include BMI as a covariate, but (4) sensitivity analyses including BMI as covariate were conducted. The sensitivity analyses handled BMI as missing at random, although this may not be fully accurate.

SAS Enterprise Guide 7.15 was used, and a significance level of 5% was applied. Validation of the programming was performed; smaller programs (3 to 20 lines of coding) were reviewed, and longer programs were double coded by an independent statistical programmer. This study is reported as per the Strengthening the Reporting of Observational Studies in Epidemiology (STROBE) guideline (see S1 Checklist). The protocol was developed prior to data access and followed, with a few exceptions: (1) Incomes should have been standardized to 2015-year level, but when grouped in quartiles, stratified on calendar year, standardization was not needed. (2) ATC codes were included to identify comorbidity, although not described in the protocol. Utilizing both ICD-10 and ATC codes more women with the given comorbidities will be identified. (3) Comedication should have been identified 1 year prior to LMP but was changed to comedication during the relevant exposure time window, since it is more likely that comedication during pregnancy will affect the outcomes of malformation and stillbirth and therefore act as confounders.

## Results

The final cohorts included up to 2,132,163 pregnancies. Among these, 1,512 and 1,668 were duloxetine exposed in the analyses of malformation and stillbirth, respectively. See flow chart in Fig 1. Up to 80,760, 64,594, and 7,699 events of major malformation, minor malformation, and stillbirth, respectively, were included in the analyses. Tables 1 and 2 show baseline characteristics for the analyses of malformation and stillbirth, respectively (Tables F and G in S1 Tables show baseline characteristics for all covariates). Table H in S1 Tables shows number of events per thousand pregnancies. Missing values ranged from 0.4% to 3.5% for household income, education, smoking, and birth order. BMI had 6.4% missing values.

For major malformations, all odds ratio (OR) point estimates based on unadjusted, adjusted, or PS-matched analyses across all 4 comparator groups are centered around 1, suggesting no increased risk (Fig 2). The PS-matched analyses for duloxetine exposed compared with duloxetine nonexposed yielded an OR of 0.98 (95% confidence interval [CI] 0.74 to 1.30,

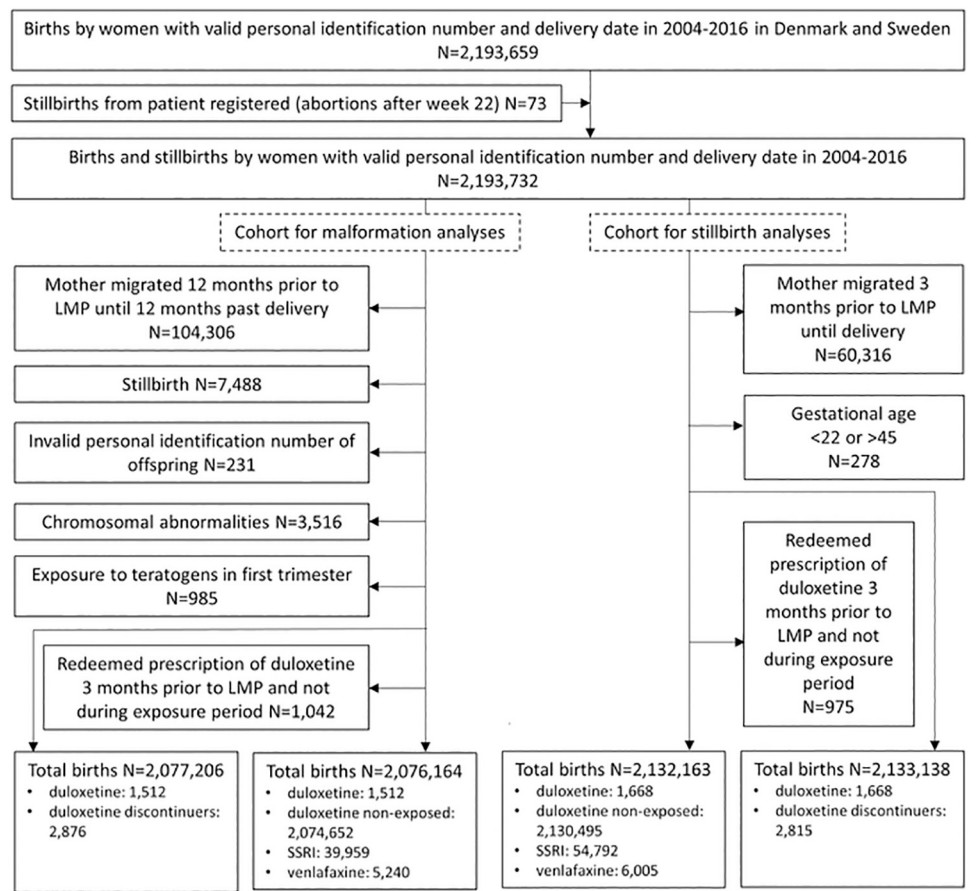

**Fig 1. Flow chart for the cohorts used to analyze malformations and stillbirth.** The 73 stillbirths identified in the patient registers were all registered as spontaneous abortions after week 22. Exposure time window for malformations: from LMP to 90 days after LMP. Exposure time window for stillbirth: from LMP to end of pregnancy. LMP, last menstrual period; SSRI, selective serotonin reuptake inhibitor.

$p$ = 0.909); for SSRI exposed, 1.07 (95% CI 0.78 to 1.46, $p$ = 0.688); for venlafaxine exposed, 0.95 (95% CI 0.66 to 1.36, $p$ = 0.783); and for duloxetine discontinuers, 0.80 (95% CI 0.56 to 1.14, $p$ = 0.213). The sensitivity analyses for major malformation the OR and 95% CI also centered around 1 and suggested no increased risk (Table I in S1 Tables).

For minor malformations, the unadjusted analysis of duloxetine exposed compared with duloxetine nonexposed showed an increased risk. However, in the adjusted and PS-matched analyses, the risk was lower and showed no statistically significant increase. When compared with SSRI exposed, venlafaxine exposed, and duloxetine discontinuers, some point estimates indicate an increased risk for minor malformations for duloxetine exposed; however, the wide CIs suggest great uncertainty (Fig 2). The PS-matched analysis for duloxetine exposed compared with duloxetine nonexposed was OR 1.09 (95% CI 0.82 to 1.45, $p$ = 0.570); for SSRI exposed, 1.39 (95% CI 1.00 to 1.94, $p$ = 0.048); for venlafaxine exposed, 1.20 (95% CI 0.82 to 1.76, $p$ = 0.337); and for duloxetine discontinuers, 1.11 (95% CI 0.77 to 1.60, $p$ = 0.574). The sensitivity analyses for minor malformation also suggested no increased risk (Table J in S1 Tables).

For the individual major malformation subtypes, all analyses were associated with large statistical uncertainty. Some point estimates suggested increased risk but were inconclusive.

**Table 1. Baseline characteristics for the analyses of malformation.**

| Variable | Value | Duloxetine Before matching n = 1,512 | Dulox. vs nonexposed — Before matching: Duloxetine nonexposed n = 2,074,652 | Std mean diff. | After matching: Duloxetine n = 1,438 | Duloxetine nonexposed n = 5,751 | Std mean diff. | Dulox. vs SSRI — Before matching: SSRI n = 39,959 | Std mean diff. | After matching: Duloxetine n = 1,437 | SSRI n = 2,874 | Std mean diff. | Dulox. vs venlafaxine — Before matching: Venlafaxine n = 5,240 | Std mean diff | After matching: Duloxetine n = 1,429 | Venlafaxine n = 1,429 | Std mean diff | Dulox. vs discontinuers — Before matching: Duloxetine discontinuers n = 2,876 | Std mean diff | After matching: Duloxetine n = 1,434 | Discontinuers n = 1,435 | Std mean diff |
|---|---|---|---|---|---|---|---|---|---|---|---|---|---|---|---|---|---|---|---|---|---|---|
| Age, continuous | Mean, y | 31 (27;35) | 30 (27;34) | 0.15 | 30.7 (27.2;35.0) | 30.8 (26.8;34.7) | 0.04 | 31 (27;34) | 0.08 | 30.7 (27.2;35.0) | 30.4 (26.6;34.5) | 0.08 | 31 (26;35) | 0.09 | 30.7 (27.2;35.0) | 30.4 (26.3;34.6) | 0.07 | 30 (27;34) | 0.11 | 30.7 (27.2;35.0) | 30.5 (26.7;34.7) | 0.04 |
| Age, grouped | 18–24 y | 231 (15.3%) | 323,541 (15.6%) | 0.16 | 222 (15.4%) | 888 (15.4%) | 0.05 | 6,399 (16.0%) | 0.06 | 222 (15.4%) | 469 (16.3%) | 0.04 | 947 (18.1%) | 0.09 | 222 (15.5%) | 240 (16.8%) | 0.03 | 470 (16.3%) | 0.07 | 222 (15.5%) | 230 (16.0%) | 0.05 |
|  | 25–29 y | 447 (29.6%) | 684,195 (33.0%) |  | 429 (29.8%) | 1,659 (28.8%) |  | 11,740 (29.4%) |  | 428 (29.8%) | 887 (30.9%) |  | 1,488 (28.4%) |  | 424 (29.7%) | 429 (30.0%) |  | 914 (31.8%) |  | 427 (29.8%) | 435 (30.3%) |  |
|  | 30–34 y | 379 (25.1%) | 591,115 (28.5%) |  | 359 (25.0%) | 1,532 (26.6%) |  | 10,855 (27.2%) |  | 359 (25.0%) | 704 (24.5%) |  | 1,359 (25.9%) |  | 356 (24.9%) | 364 (25.5%) |  | 710 (24.7%) |  | 359 (25.0%) | 336 (23.4%) |  |
|  | 35–60 y | 455 (30.1%) | 475,801 (22.9%) |  | 428 (29.8%) | 1,672 (29.1%) |  | 10,965 (27.4%) |  | 428 (29.8%) | 814 (28.3%) |  | 1,446 (27.6%) |  | 427 (29.9%) | 396 (27.7%) |  | 782 (27.2%) |  | 426 (29.7%) | 434 (30.2%) |  |
| Household income | Quartile1 | 569 (37.8%) | 458,644 (22.2%) | 0.41 | 549 (38.2%) | 2,189 (38.1%) | 0.00 | 12,032 (30.3%) | 0.19 | 548 (38.1%) | 1,115 (38.8%) | 0.03 | 1,895 (36.4%) | 0.05 | 545 (38.1%) | 569 (39.8%) | 0.05 | 1,051 (36.7%) | 0.05 | 549 (38.3%) | 560 (39.0%) | 0.05 |
|  | Quartile2 | 391 (25.9%) | 514,943 (25.0%) |  | 370 (25.7%) | 1,481 (25.8%) |  | 10,306 (26.0%) |  | 370 (25.7%) | 757 (26.3%) |  | 1,418 (27.3%) |  | 367 (25.7%) | 369 (25.8%) |  | 815 (28.5%) |  | 367 (25.6%) | 346 (24.1%) |  |
|  | Quartile3 | 317 (21.0%) | 547,822 (26.6%) |  | 300 (20.9%) | 1,222 (21.2%) |  | 9,760 (24.6%) |  | 300 (20.9%) | 610 (21.2%) |  | 1,106 (21.3%) |  | 298 (20.9%) | 280 (19.6%) |  | 597 (20.9%) |  | 300 (20.9%) | 317 (22.1%) |  |
|  | Quartile4 | 230 (15.3%) | 540,500 (26.2%) |  | 219 (15.2%) | 859 (14.9%) |  | 7,557 (19.1%) |  | 219 (15.2%) | 392 (13.6%) |  | 784 (15.1%) |  | 219 (15.3%) | 211 (14.8%) |  | 399 (13.9%) |  | 218 (15.2%) | 212 (14.8%) |  |
| Education | <11 y | 341 (22.7%) | 252,162 (12.3%) | 0.40 | 327 (22.7%) | 1,462 (25.4%) | 0.05 | 7,390 (18.6%) | 0.19 | 327 (22.8%) | 685 (23.8%) | 0.03 | 1,221 (23.5%) | 0.06 | 324 (22.7%) | 309 (21.6%) | 0.03 | 714 (25.0%) | 0.06 | 325 (22.7%) | 301 (21.0%) | 0.05 |
|  | 11–15 y | 806 (53.7%) | 1,000,210 (48.8%) |  | 771 (53.6%) | 3,010 (52.3%) |  | 19,650 (49.5%) |  | 770 (53.6%) | 1,540 (53.6%) |  | 2,641 (50.9%) |  | 765 (53.5%) | 774 (54.2%) |  | 1,452 (50.8%) |  | 770 (53.7%) | 791 (55.1%) |  |
|  | >16 y | 355 (23.6%) | 797,606 (38.9%) |  | 340 (23.6%) | 1,279 (22.2%) |  | 12,630 (31.8%) |  | 340 (23.7%) | 649 (22.6%) |  | 1,326 (25.6%) |  | 340 (23.8%) | 346 (24.2%) |  | 692 (24.2%) |  | 339 (23.6%) | 343 (23.9%) |  |
| Smoking | Yes | 301 (20.7%) | 179,398 (9.0%) | 0.34 | 298 (20.7%) | 1,339 (23.3%) | −0.06 | 6,640 (17.2%) | 0.09 | 297 (20.7%) | 570 (19.8%) | 0.02 | 1,303 (25.6%) | −0.12 | 296 (20.7%) | 272 (19.0%) | 0.04 | 578 (20.9%) | −0.01 | 296 (20.6%) | 285 (19.9%) | 0.02 |
| Data source, Sweden | Yes | 1,010 (66.8%) | 1,324,668 (63.9%) | 0.06 | 959 (66.7%) | 3,900 (67.8%) | −0.02 | 2,5975 (65.0%) | 0.03 | 958 (66.7%) | 1,881 (65.4%) | 0.03 | 2,948 (56.3%) | 0.22 | 950 (66.5%) | 960 (67.2%) | −0.01 | 1,821 (63.3%) | 0.07 | 955 (66.6%) | 950 (66.2%) | 0.01 |
| Previous stillbirth | Yes | 6 (0.4%) | 10,401 (0.5%) | −0.02 | 6 (0.4%) | 76 (1.3%) | −0.10 | 250 (0.6%) | 0.02 | 6 (0.4%) | 8 (0.3%) | 0.02 | 22 (0.4%) | 0.00 | 6 (0.4%) | 6 (0.4%) | 0.00 | 17 (0.6%) | −0.03 | 6 (0.4%) | 5 (0.3%) | 0.01 |
| Depression | Yes | 517 (34.2%) | 39,131 (1.9%) | 0.93 | 486 (33.8%) | 2,082 (36.2%) | −0.05 | 7,326 (18.3%) | 0.37 | 485 (33.8%) | 1002 (34.9%) | −0.02 | 1,324 (25.3%) | 0.20 | 478 (33.4%) | 509 (35.6%) | −0.05 | 812 (28.2%) | 0.13 | 482 (33.6%) | 477 (33.2%) | 0.01 |
| Anxiety or phobia | Yes | 193 (12.8%) | 19,287 (0.9%) | 0.48 | 181 (12.6%) | 769 (13.4%) | −0.02 | 4,022 (10.1%) | 0.08 | 181 (12.6%) | 341 (11.9%) | 0.02 | 508 (9.7%) | 0.10 | 177 (12.4%) | 202 (14.1%) | −0.05 | 327 (11.4%) | 0.04 | 180 (12.6%) | 162 (11.3%) | 0.04 |
| Severe stress reaction | Yes | 171 (11.3%) | 21,233 (1.0%) | 0.44 | 153 (10.6%) | 576 (10.0%) | 0.02 | 2,010 (5.0%) | 0.23 | 153 (10.6%) | 296 (10.3%) | 0.01 | 353 (6.7%) | 0.16 | 149 (10.4%) | 158 (11.1%) | −0.02 | 278 (9.7%) | 0.05 | 151 (10.5%) | 137 (9.5%) | 0.03 |
| Stress urinary incontinence | Yes | <5 (0%) | <5 (0%) | 0.00 | <5 (0%) | <5 (0%) | 0.00 | <5 (0%) | 0.00 | <5 (0%) | <5 (0%) | 0.00 | <5 (0%) | 0.00 | <5 (0%) | <5 (0%) | 0.00 | <5 (0%) | 0.00 | <5 (0%) | <5 (0%) | 0.00 |
| SSRI comedication | Yes | <5 (0%) | 39,959 (1.9%) | −0.20 | <5 (0%) | 923 (16.0%) | −0.62 | N/A | N/A | N/A | N/A | N/A | 710 (13.5%) | −0.56 | <5 (0%) | 185 (12.9%) | −0.55 | 579 (20.1%) | −0.71 | <5 (0%) | 302 (21.0%) | −0.73 |
| Venlafaxine comedication | Yes | <5 (0%) | 5,240 (0.3%) | −0.07 | <5 (0%) | 173 (3.0%) | −0.25 | 710 (1.8%) | −0.19 | <5 (0%) | 71 (2.5%) | −0.23 | N/A | N/A | N/A | N/A | N/A | 88 (3.1%) | −0.25 | <5 (0%) | 49 (3.4%) | −0.27 |

Previous stillbirth, depression, anxiety or phobia, and severe stress reaction were defined as diagnoses up to 5 years prior to LMP. PS-matched models were based on covariates covering comorbidity (up to 5 years prior to LMP), comedication (during the relevant time period), hospital contacts, education, and income. For the complete list for the individual analyses, see Table E in S1 Tables. SSRI and venlafaxine comedication were not part of the confounder variables.

LMP, last menstrual period; PS, propensity score; SSRI, selective serotonin reuptake inhibitor; Std mean diff, standardized mean difference; y, years.

Results for the major malformation subtypes are found in the Supporting information (both main and sensitivity analyses; Tables K-V in S1 Tables). For cardiac malformations, PS-matched analysis for duloxetine exposed compared with duloxetine nonexposed was OR 1.01 (95% CI 0.64 to 1.60, $p = 0.962$); for SSRI exposed, 0.79 (95% CI 0.49 to 1.29, $p = 0.344$); for venlafaxine exposed, 0.78 (95% CI 0.44 to 1.38, $p = 0.388$); and for duloxetine discontinuers, 0.92 (95% CI 0.51 to 1.63, $p = 0.768$). It is of note that a statistically significant increased risk of "other anomalies and syndromes" was found when duloxetine-exposed women were compared with SSRI exposed (PS-matched analyses of duloxetine versus SSRI: OR 2.43 [95% CI 1.10 to 5.38, $p = 0.028$]). See Table B in S1 Tables for the full list of other anomalies and syndromes, but it includes, e.g., craniosynostosis, situs inversus, and fetal alcohol syndrome.

For stillbirths, the analyses suggested no increased risk for duloxetine exposed across all comparison groups, although the CIs were wide (Fig 3). PS-matched analysis for duloxetine

**Table 2. Baseline characteristics for the analyses of stillbirth.**

| Variable | Value | Duloxetine Before matching n = 1,668 | Duloxetine vs. duloxetine nonexposed | | | | | Duloxetine vs. SSRI | | | | | Duloxetine vs. venlafaxine | | | | | Duloxetine vs. duloxetine discontinuers | | | | |
|---|---|---|---|---|---|---|---|---|---|---|---|---|---|---|---|---|---|---|---|---|---|---|
| | | | Before matching | | After matching | | | Before matching | | After matching | | | Before matching | | After matching | | | Before matching | | After matching | | |
| | | | Duloxetine nonexposed n = 2,130,495 | Std mean diff. | Duloxetine n = 1,581 | Duloxetine nonexposed n = 6,324 | Std mean diff. | SSRI n = 54,792 | Std mean diff. | Duloxetine n = 1,585 | SSRI n = 3,170 | Std mean diff. | Venlafaxine n = 6,005 | Std mean diff | Duloxetine n = 1,580 | Venlafaxine n = 1,580 | Std mean diff | Duloxetine discontinuers n = 2,815 | Std mean diff | Duloxetine n = 1,559 | Discontinuers n = 1,561 | Std mean diff |
| Age, continuous | Mean, y | 31.0 (27.3;35.1) | 30.2 (26.7;33.6) | 0.18 | 30.9 (27.3;35.1) | 30.9 (26.6;35.0) | 0.05 | 30.7 (26.8;34.4) | | 30.9 (27.3;35.1) | 30.7 (26.9;34.9) | 0.05 | 30.6 (26.6;34.6) | | 30.9 (27.3;35.1) | 30.6 (26.9;35.1) | 0.02 | 30.3 (26.6;34.3) | | 30.9 (27.3;35.1) | 30.8 (27.0;34.7) | 0.04 |
| Age, grouped | 18–24 y | 245 (14.7%) | 335,954 (15.8%) | 0.16 | 235 (14.9%) | 1,087 (17.2%) | 0.07 | 8,706 (15.9%) | | 236 (14.9%) | 464 (14.6%) | 0.05 | 1,038 (17.3%) | | 235 (14.9%) | 227 (14.4%) | 0.03 | 470 (16.7%) | | 236 (15.1%) | 231 (14.8%) | 0.00 |
| | 25–29 y | 485 (29.1%) | 702,496 (33.0%) | | 466 (29.5%) | 1,723 (27.2%) | | 16,148 (29.5%) | | 467 (29.5%) | 973 (30.7%) | | 1,720 (28.6%) | | 466 (29.5%) | 477 (30.2%) | | 893 (31.7%) | | 460 (29.5%) | 466 (29.9%) | |
| | 30–34 y | 426 (25.5%) | 604,784 (28.4%) | | 398 (25.2%) | 1,529 (24.2%) | | 14,876 (27.1%) | | 400 (25.2%) | 769 (24.3%) | | 1,566 (26.1%) | | 398 (25.2%) | 390 (24.7%) | | 692 (24.6%) | | 391 (25.1%) | 395 (25.3%) | |
| | 35–60 y | 512 (30.7%) | 487,539 (22.9%) | | 482 (30.5%) | 1,985 (31.4%) | | 15,067 (27.5%) | | 482 (30.4%) | 964 (30.4%) | | 1,684 (28.0%) | | 481 (30.4%) | 486 (30.8%) | | 762 (27.1%) | | 472 (30.3%) | 467 (30.0%) | |
| Household income | Quartile1 | 627 (37.7%) | 481,338 (22.8%) | 0.39 | 601 (38.0%) | 2,371 (37.5%) | 0.04 | 16,538 (30.4%) | | 605 (38.2%) | 1,232 (38.9%) | 0.03 | 2,172 (36.4%) | | 603 (38.2%) | 606 (38.4%) | 0.06 | 1,031 (36.8%) | | 595 (38.2%) | 603 (38.7%) | 0.03 |
| | Quartile2 | 427 (25.7%) | 529,514 (25.1%) | | 402 (25.4%) | 1,544 (24.4%) | | 14,121 (26.0%) | | 402 (25.4%) | 793 (25.0%) | | 1,617 (27.1%) | | 401 (25.4%) | 423 (26.8%) | | 806 (28.8%) | | 396 (25.4%) | 393 (25.2%) | |
| | Quartile3 | 353 (21.2%) | 554,446 (26.2%) | | 334 (21.1%) | 1,406 (22.2%) | | 13,320 (24.5%) | | 334 (21.1%) | 677 (21.4%) | | 1,261 (21.1%) | | 333 (21.1%) | 331 (20.9%) | | 578 (20.6%) | | 326 (20.9%) | 327 (21.0%) | |
| | Quartile4 | 255 (15.3%) | 547,490 (25.9%) | | 244 (15.4%) | 1,003 (15.9%) | | 10,383 (19.1%) | | 244 (15.4%) | 468 (14.8%) | | 913 (15.3%) | | 243 (15.4%) | 220 (13.9%) | | 388 (13.8%) | | 242 (15.5%) | 236 (15.1%) | |
| Education | <11 y | 374 (22.6%) | 262,602 (12.5%) | 0.35 | 601 (38.0%) | 1,617 (25.6%) | 0.07 | 10,068 (18.5%) | | 360 (22.7%) | 707 (22.3%) | 0.03 | 1,386 (23.3%) | | 357 (22.6%) | 333 (21.1%) | 0.05 | 708 (25.3%) | | 355 (22.8%) | 348 (22.3%) | 0.03 |
| | 11–15 y | 878 (53.1%) | 1,020,571 (48.6%) | | 402 (25.4%) | 3,191 (50.5%) | | 26,933 (49.6%) | | 840 (53.0%) | 1,703 (53.7%) | | 2,995 (50.5%) | | 838 (53.0%) | 868 (54.9%) | | 1,425 (50.9%) | | 824 (52.9%) | 842 (54.0%) | |
| | >16 y | 402 (24.3%) | 816,010 (38.9%) | | 334 (21.1%) | 1,516 (24.0%) | | 17,306 (31.9%) | | 385 (24.3%) | 760 (24.0%) | | 1,555 (26.2%) | | 385 (24.4%) | 379 (24.0%) | | 667 (23.8%) | | 380 (24.4%) | 369 (23.7%) | |
| Smoking | Yes | 336 (20.9%) | 183,149 (8.9%) | 0.34 | 332 (21.0%) | 1,447 (22.9%) | −0.05 | 9,011 (17.0%) | | 332 (20.9%) | 661 (20.9%) | 0.00 | 1,469 (25.3%) | | 331 (20.9%) | 321 (20.3%) | 0.02 | 560 (20.7%) | | 324 (20.8%) | 316 (20.3%) | 0.01 |
| Data source, Sweden | Yes | 1,120 (67.1%) | 1,360,775 (63.9%) | 0.07 | 1,059 (67.0%) | 4,293 (67.9%) | −0.02 | 36,616 (66.8%) | | 1,063 (67.1%) | 2,086 (65.8%) | 0.03 | 3,414 (56.8%) | | 1,058 (67.0%) | 1,084 (68.6%) | −0.04 | 1,770 (62.8%) | | 1,040 (66.7%) | 1,007 (64.6%) | 0.04 |
| Previous stillbirth | Yes | 6 (0.4%) | 10,898 (0.5%) | −0.02 | 6 (0.4%) | 90 (1.4%) | −0.11 | 339 (0.6%) | | 6 (0.4%) | 10 (0.3%) | 0.01 | 29 (0.5%) | | 6 (0.4%) | 8 (0.5%) | −0.02 | 19 (0.7%) | | 6 (0.4%) | 5 (0.3%) | 0.01 |
| Depression | Yes | 1,090 (65.3%) | 2,091,153 (98.1%) | 0.94 | 541 (34.2%) | 2,342 (37.0%) | −0.06 | 9,984 (18.2%) | | 545 (34.4%) | 1,129 (35.6%) | −0.03 | 1,529 (25.4%) | | 540 (34.2%) | 572 (36.2%) | −0.04 | 775 (27.5%) | | 524 (33.6%) | 525 (33.7%) | 0.00 |
| Anxiety or phobia | Yes | 217 (13.0%) | 19,510 (0.9%) | 0.49 | 202 (12.8%) | 811 (12.8%) | −0.00 | 5,348 (9.8%) | | 205 (12.9%) | 442 (13.9%) | −0.03 | 575 (9.6%) | | 202 (12.8%) | 210 (13.3%) | −0.02 | 312 (11.1%) | | 194 (12.4%) | 50 (3.2%) | 0.01 |
| Severe stress reaction | Yes | 203 (12.2%) | 21,580 (1.0%) | 0.46 | 181 (11.4%) | 679 (10.7%) | 0.02 | 2,815 (5.1%) | | 184 (11.6%) | 363 (11.5%) | 0.00 | 399 (6.6%) | | 180 (11.4%) | 176 (11.1%) | 0.01 | 260 (9.2%) | | 173 (11.1%) | 175 (11.2%) | 0.00 |
| Stress urinary incontinence | Yes | <5 (0%) | <5 (0%) | 0.00 | <5 (0%) | <5 (0%) | 0.00 | <5 (0%) | 0.00 | <5 (0%) | <5 (0%) | 0.00 | <5 (0%) | 0.00 | <5 (0%) | <5 (0%) | 0.00 | <5 (0%) | 0.00 | <5 (0%) | <5 (0%) | 0.00 |
| SSRI comedication | Yes | <5 (0%) | 54,797 (2.6%) | −0.23 | <5 (0%) | 1,219 (19.3%) | −0.69 | N/A | N/A | N/A | N/A | N/A | 1,255 (20.9%) | −0.73 | <5 (0%) | 322 (20.4%) | −0.72 | 778 (27.6%) | −0.87 | <5 (0%) | 472 (30.3%) | −0.93 |
| Venlafaxine comedication | Yes | <5 (0%) | 6,008 (0.3%) | −0.08 | <5 (0%) | 249 (3.9%) | −0.29 | 1,255 (2.3%) | −0.22 | <5 (0%) | 107 (3.4%) | −0.26 | N/A | N/A | N/A | N/A | N/A | 95 (3.4%) | −0.26 | <5 (0%) | 59 (3.8%) | −0.28 |

Previous stillbirth, depression, anxiety or phobia, and severe stress reaction were defined as diagnoses up to 5 years prior to LMP. PS-matched models were based on covariates covering comorbidity (up to 5 years prior to LMP), comedication (during the relevant time period), hospital contacts, education, and income. For the complete list for the individual analyses, see Table E in S1 Tables. SSRI and venlafaxine comedication were not part of the confounder variables.

LMP, last menstrual period; PS, propensity score; SSRI, selective serotonin reuptake inhibitor; Std mean diff, standardized mean difference; y, years.

exposed compared with duloxetine nonexposed was 1.18 (95% CI 0.43 to 3.19, *p* = 0.749); for SSRI exposed, 0.63 (95% CI 0.23 to 1.71, *p* = 0.359); for venlafaxine exposed, 0.42 (95% CI 0.15 to 1.18, *p* = 0.100); and for duloxetine discontinuers, 0.83 (95% CI 0.25 to 2.73, *p* = 0.763). The sensitivity analyses for stillbirth also suggested no increased risk (Table W in S1 Tables).

## Discussion

This observational study with nationwide register data from Sweden and Denmark found no increased risk of congenital minor or major malformations or stillbirths among women exposed to duloxetine during pregnancy. For minor malformations, there was some tendency for an increased risk, but estimates had wide CIs and the tendency decreased when duloxetine exposed were compared with venlafaxine exposed or duloxetine discontinuers, as well as in

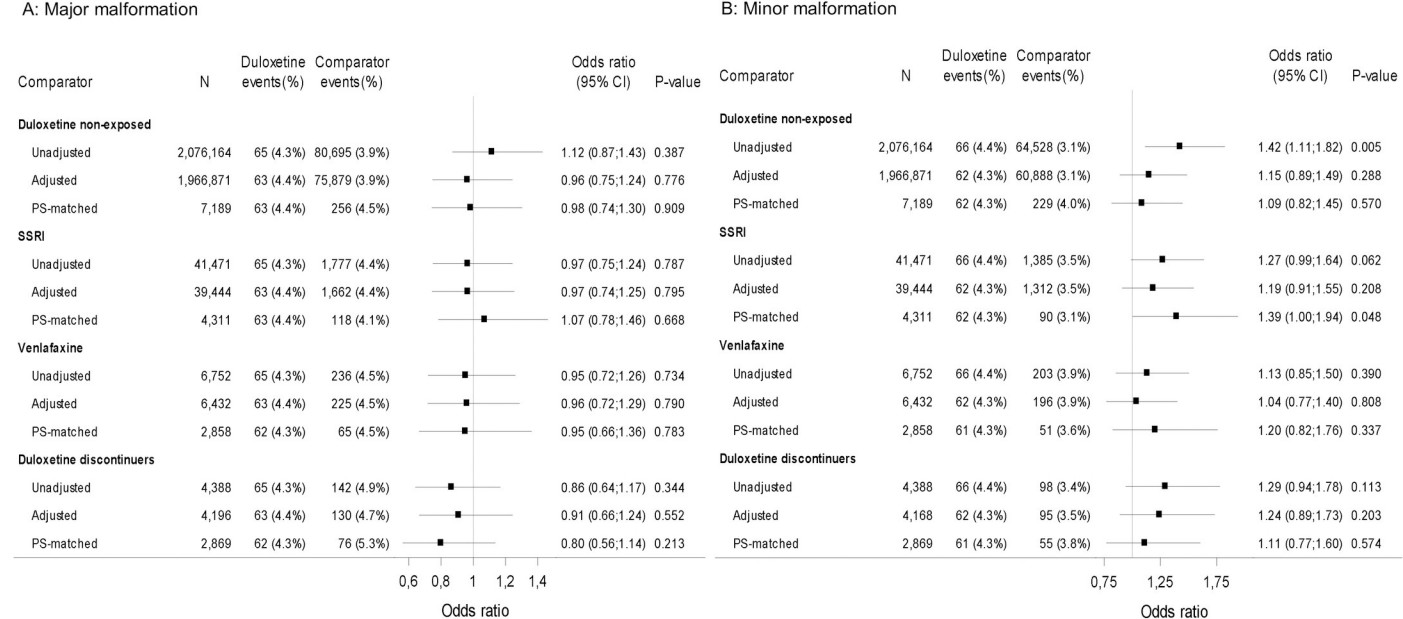

**Fig 2. Risk of major and minor malformation.** Duloxetine vs 4 comparators. Odds ratio for major or minor malformations for duloxetine vs comparator. Exposure definition: ≥1 redeemed prescription. Adjusted and PS-matched models were based on covariates covering comorbidity (up to 5 years prior to LMP), comedication (90 days prior to LMP to end of the relevant exposure time window), hospital contacts, education, and income. For the complete list of covariates for the individual analyses, see Table E in S1 Tables. CI, Wald 95% confidence intervals; LMP, last menstrual period; N, number of observations in analyses; PS-matched, propensity score–matched analyses based on conditional logistic regression; SSRI, selective serotonin reuptake inhibitor.

adjusted and PS-matched analyses. Confounding by indication or other unmeasured confounding may explain the risk.

## Interpretation

The results of the present study are in line with previous studies [31,32] and case reports [26,52,53] finding no increased risk for malformations associated with duloxetine exposure. A review from 2015 concluded that there was insufficient data on duloxetine to draw definitive conclusions about its safety in pregnancy [54]. A review from 2016 concluded that women exposed to duloxetine during the first trimester (*n* = 668) had no increased risk of congenital malformations (OR 0.80, 95% CI 0.46 to 1.29) [33]. The present study corroborates these findings based on a considerably larger group of duloxetine-exposed women (*n* = 1,512) while addressing more potential confounding factors and including PS-matched analyses and sensitivity analyses. Huybrechts and colleagues [32] did a recent study in the US and found no increased risk for malformation overall, but a small increased risk of cardiac malformation. Although no increased risk of cardiac malformations was found in the present study, it must be noted that there was an increased risk of "other anomalies and syndromes" (See Table B in S1 Tables for the full list of other anomalies and syndromes, but it includes, e.g., craniosynostosis, situs inversus, and fetal alcohol syndrome). This must be interpreted with caution based on the low number of cases (*n* = 14) and the wide CIs. Since we find no clear pharmacological mechanism explaining the association, the result is interpreted to be a chance finding.

A small Swedish study from 2007 investigating SNRI/NRIs (not including duloxetine) found no increased risk for stillbirth when compared to the background population or to women exposed to SSRI during pregnancy [55]. The present study, with a bigger population and more recent data, supports this finding.

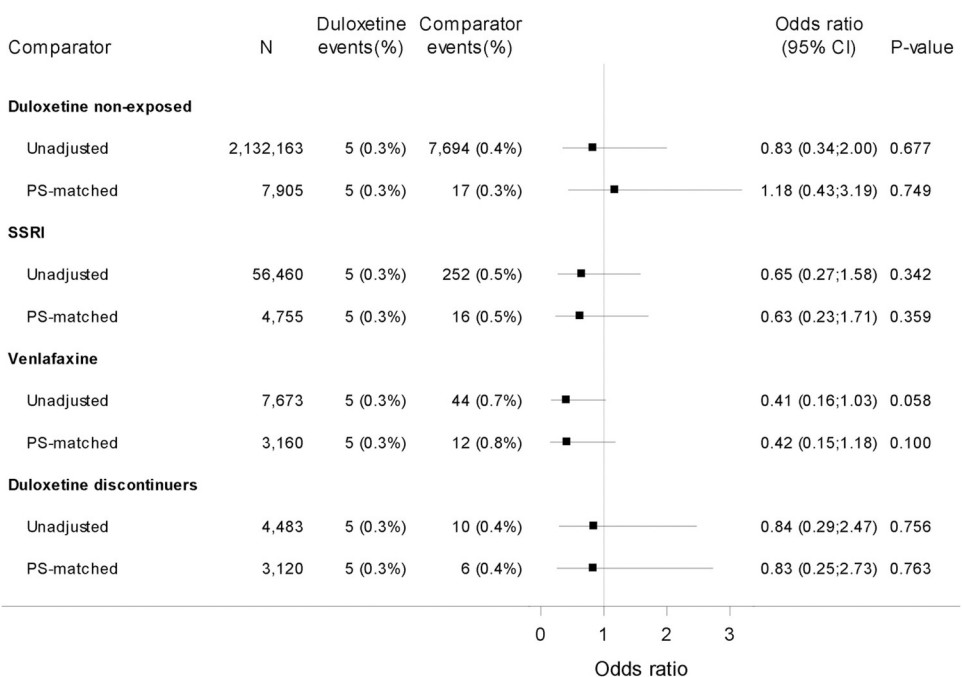

**Fig 3. Risk of stillbirth.** Duloxetine vs 4 comparators. Odds ratio for stillbirth for duloxetine vs comparator. Exposure definition: ≥1 redeemed prescription. PS-matched models were based on covariates covering comorbidity (up to 5 years prior to LMP), comedication (90 days prior to LMP to end of the relevant exposure time window), hospital contacts, education, and income. For the complete list of covariates for the individual analyses, see Table E in S1 Tables. CI, Wald 95% confidence intervals; LMP, last menstrual period; N, number of observations in analyses; PS-matched, propensity score–matched analyses based on conditional logistic regression; SSRI, selective serotonin reuptake inhibitor.

## Strengths and limitations

Due to the nationwide coverage, high validity, and completeness of the included registers, the risk of selection (sampling, allocation, and lost to follow-up) bias was minimal. There was no risk of recall bias. The quality of malformation diagnoses was validated in Danish data and was found to have a predictive value of 88%, with a completeness of 90% [56]. Diagnoses of heart defects were found to have a positive predictive value of 98.4% [57]. For the present study, differential misclassification depending on exposure status was not suspected. Information on malformation among abortions and stillbirths was not available. This can lead to an underestimation of the risk if duloxetine exposure is associated with malformations leading to these outcomes. Information on LMP was precise as it was based on the mother's self-report and 2 subsequent ultrasounds in the first and second trimester.

The primary exposure definition was the redemption of a single prescription. Although the medication has been prescribed, dispensed, redeemed, and paid for, there is a probability that the patient did not ingest the drug. Sensitivity analyses were performed with a stricter definition of exposure (>1 redeemed prescription) under the assumption that redeeming multiple prescriptions increases the likelihood that the medication was taken. These sensitivity analyses did not change the overall results. Information on drug exposure during hospitalization was unavailable. This might have led to misclassification of exposure as exposure is based on redeemed prescriptions from community pharmacies. However, we assume that women hospitalized because of depression also redeem prescriptions of antidepression medication. Also, it is expected that most women with the indication for antidepressants are treated outside the

hospital since only 1.6% of the total duloxetine use in the time period was administrated in hospitals in Denmark [58]. Important unavailable potential confounders were alcohol, illicit drug use, and poor adherence to folic acid supplementation during pregnancy [59]. Women with depressive disorder are more likely to smoke, use alcohol or other substances, and, in general, not to adhere to recommended health behavior during pregnancy. This may confound the association between duloxetine and pregnancy outcomes and increased risk of malformations may erroneously be attributed to duloxetine [60,61]. Socioeconomic status (education and income) and smoking were included as covariates. Due to the concern of unmeasured confoundings (e.g., alcohol and illicit drug use) and confounding by indication, SSRI-exposed and venlafaxine-exposed women were used as comparators as they are expected to have similar health behavior as duloxetine-exposed women.

Indication for duloxetine was not available as a recorded covariate. In addition to depression, duloxetine is indicated for the treatment of neuropathy, anxiety, severe stress reaction, and stress urinary incontinence. In the analyzed cohort of pregnant women, the prevalence of these indications is, however, expected to be very low. Depression severity, which might also lead to confounding, and a direct measurement for depression severity was unavailable. This potential confounding was addressed by including diagnoses of neuropathy, anxiety, severe stress reaction, and stress urinary incontinence as covariates, as well as covariates that describe depression severity (depression diagnosis recorded at hospital contact, psychiatric hospitalization, and outpatient visits).

In the analyses of major malformation subtypes and stillbirth, adjusted analyses could not be consistently performed due to the low number of outcome events despite the cohort including more than 2 million births. As opposed to adjusted analyses, the PS-matched analyses were possible despite few outcome events.

The study included all registered pregnancies resulting in a birth (live or stillbirth). We believe that the results have a high external validity especially applicable to other western European countries with free and universal healthcare, where treatment regimens and population characteristics are comparable, as well as to the US, where indications and treatment guidelines are similar to the studied population. Despite the large cohort, the number of women exposed to duloxetine during pregnancy was limited, and future studies should focus on analyzing larger cohorts and additional safety outcomes (e.g., preterm birth, abortions, and SGA). Improved measurements of exposure, outcome, and covariates could also yield more precise estimates. In addition, information on duloxetine indication and depression severity could add relevant information.

## Conclusions

Based on this observational register-based nationwide study with data from Sweden and Denmark, no increased risk of major and minor congenital malformations or stillbirths was associated with exposure to duloxetine during pregnancy.

## Supporting information

**S1 STROBE Checklist. STROBE Statement—Checklist of items that should be included in reports of cohort studies.**
(PDF)

**S1 Protocol. Observational study to assess maternal and fetal outcomes following exposure to duloxetine: Denmark and Sweden National Pregnancy Registry Study.**
(PDF)

**S1 Tables.** **Table A.** Minor malformation ICD-10 codes. **Table B.** ICD-10 codes for major malformation subtypes. **Table C.** ICD-10, ATC codes, and time periods for comorbidity. **Table D.** ATC codes for comedication. **Table E.** Covariates in the models of the primary analyses (the figures shown in the manuscript). **Table F.** Baseline table for all covariates, major and minor malformation analyses. Before and after propensity score matching. **Table G.** Baseline table for all covariates, stillbirth analyses. Before and after propensity score matching. **Table H.** Number of events per thousand pregnancies (95% Wald confidence intervals). **Table I.** Major malformation, sensitivity analyses. **Table J.** Minor malformation, sensitivity analyses. **Table K.** Malformation subtype: Heart defect. **Table L.** Malformation subtype: Digestive system. **Table M.** Malformation subtype: Ear, face, and neck. **Table N.** Malformation subtype: Eye. **Table O.** Malformation subtype: Genitals. **Table P.** Malformation subtype: Abdominal wall. **Table Q.** Malformation subtype: Limb. **Table R.** Malformation subtype: Nervous system. **Table S.** Malformation subtype: Orofacial clefts. **Table T.** Malformation subtype: Respiratory. **Table U.** Malformation subtype: Urinary tract. **Table V.** Malformation subtype: Other anomalies/syndromes. **Table W.** Stillbirth, sensitivity analyses.
(DOCX)

## Acknowledgments

### Funding

This publication is based on a post-authorization safety study (PASS) on duloxetine requested by the Food and Drug Administration (FDA). Eli Lilly and Company funded the PASS study. The funder had the opportunity to comment study design, data collection and analysis, where to publish and preparation of the manuscript, but MZA and EJS had the final decisions.

## Author Contributions

**Conceptualization:** Mikkel Zöllner Ankarfeldt, Janne Petersen, Jon Trærup Andersen, Hu Li, Espen Jimenez-Solem.

**Data curation:** Mikkel Zöllner Ankarfeldt, Janne Petersen, Thomas Fast.

**Formal analysis:** Mikkel Zöllner Ankarfeldt, Janne Petersen.

**Funding acquisition:** Hu Li, Stephen Paul Motsko.

**Methodology:** Mikkel Zöllner Ankarfeldt, Janne Petersen, Jon Trærup Andersen, Espen Jimenez-Solem.

**Project administration:** Hu Li, Stephen Paul Motsko, Simone Møller Hede.

**Resources:** Hu Li, Stephen Paul Motsko.

**Supervision:** Janne Petersen, Thomas Fast, Espen Jimenez-Solem.

**Validation:** Mikkel Zöllner Ankarfeldt, Janne Petersen.

**Visualization:** Mikkel Zöllner Ankarfeldt, Janne Petersen.

**Writing – original draft:** Mikkel Zöllner Ankarfeldt.

**Writing – review & editing:** Mikkel Zöllner Ankarfeldt, Janne Petersen, Jon Trærup Andersen, Hu Li, Stephen Paul Motsko, Thomas Fast, Simone Møller Hede, Espen Jimenez-Solem.

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
