## [Editor Report · Decision Letter 0]

30 Sep 2020

Dear Dr Ankarfeldt, 

Thank you for submitting your manuscript entitled "Exposure to duloxetine during pregnancy and risk of congenital malformations and stillbirth

A nation-wide cohort study in Denmark and Sweden" for consideration by PLOS Medicine.

Your manuscript has now been evaluated by the PLOS Medicine editorial staff as well as by an academic editor with relevant expertise and I am writing to let you know that we would like to send your submission out for external peer review.

Kind regards,

Thomas J McBride, PhD,

Senior Editor

PLOS Medicine

---

## [Decision Letter · Decision Letter 1]

3 Feb 2021

Dear Dr. Ankarfeldt,

Thank you very much for submitting your manuscript "Exposure to duloxetine during pregnancy and risk of congenital malformations and stillbirth A nation-wide cohort study in Denmark and Sweden" (PMEDICINE-D-20-04640R1) for consideration at PLOS Medicine. 

[LINK]

In light of these reviews, I am afraid that we will not be able to accept the manuscript for publication in the journal in its current form, but we would like to consider a revised version that addresses the reviewers' and editors' comments. Obviously we cannot make any decision about publication until we have seen the revised manuscript and your response, and we plan to seek re-review by one or more of the reviewers. 

We expect to receive your revised manuscript by Feb 24 2021 11:59PM. Please email us (plosmedicine@plos.org) if you have any questions or concerns.

We look forward to receiving your revised manuscript. 

Sincerely,

Raffaella

Dr Raffaella Bosurgi MSc, PhD

Executive Editor, PLOS Medicine

rbosurgi@plos.org

https://twitter.com/raffi74

Remote based in London, UK

PLOS

Comments from the reviewers:

Reviewer #1: "Exposure to duloxetine during pregnancy and risk of congenital malformations and stillbirth A nation-wide cohort study in Denmark and Sweden" seeks to investigate the possible effect of Duloxetine (prescribed as an antidepression etc.) exposure, on (major/minor) congenital infant malformation and stillbirth. The study was conducted on a cohort of all pregnancies in Sweden and Denmark, from 2004-2016. The major finding was that duloxetine exposure did not result in increased risk of malformation/stillbirth, from adjusted and propensity score-matched (cohort details in Table S6A/B) logistic regression analyses.

Concerns may remain over the relatively low number of Duloxetine events (60+ for major/minor malformation, 5 stillbirths), which would seem to make drawing definitive conclusions difficult. However, given the national-level scope of the study and lack of prior large cohort studies, this would appear to be a valuable addition to the literature. Nevertheless, some issues might be addressed:

1. For the cohort selection flowchart from Figure 1, there appears to possibly be an inconsistency. It is presented that both malformation & stillbirth cohorts begin with the same initial cohort (N=2,193,732), and that the malformation cohort has N=1,042 subjects who "redeemed prescription of duloxetine 3 months prior to LMP and not during exposure period".

However, given that the only prior exclusion criteria for the stillbirth cohort was "mother migrated 3 months prior to LMP until delivery", and that this appears a strict subset of "mother migrated 12 months prior to LMP until 12 months past delivery" for the malformation cohort (amongst other exclusion criteria for the malformation cohort), it would appear that the corresponding number of subjects excluded for "redeemed prescription of duloxetine 3 months prior to LMP and not during exposure period" for the stillbirth cohort should be not less than that for the malformation cohort (whereas it is presented as N=975). The authors might clarify if this reasoning is correct or flawed.

2. On Page 8, it is mentioned that "When fitting each model, covariates were removed, if the model could not be estimated" for adjusted/PS models. The authors might consider explaining in greater detail why certain covariates would cause a model not to be able to be estimated, and what models were affected/covariates were removed for these affected models. Details on excluded covariates might be added to Supplementary Figures S1 to S12 as appropriate.

3. On Page 9, it is mentioned that missing income would be imputed from one year prior and then one year after LMP if possible, and 1 year prior for education. Does this cover all missing data cases, and if not, what was the final imputed value for cases where no income/education/etc. data was available at all?

4. On Page 9, it is noted that four sensitivity analyses were performed. However, their charts/details were not located, despite patterns on the results of these four sensitivity analyses being referenced in the text discussion. In particular, for BMI, the main analyses in Figures 2 to 4 do not appear to mention BMI in their marker descriptions, while subanalyses from Figure S2 onwards mention BMI (grouped) in marker 3, which however does not appear to have been applied (also covered as a minor issue). The authors might consider including the details of these sensitivity analyses.

5. On Page 14, it is stated that "...adjusted analyses could not consistently be performed due to the low number of outcome events despite of the cohort including more than 2 million births. The PS matched analyses performed better with few outcome events."; it might be clarified as to what "better" in the second sentence means - better in some statistical sense, or better by the results?

Minor issues:

6. On Page 8, the relevant literature contributing to covariate selection might be referenced if possible.

7. The motivation for using SSRI-exposed and Venlafaxine exposed patients as comparators might be moved from the discussion (Page 13), to the initial description of the methodology.

8. From Figure 1, it might be clarified whether the N=73 stillbirths (abortions after week 22) were spontaneous or intentional.

9. Still on the malformation subtype analyses as presented in Supplementary Figures S1 to S12, certain values are presented as <5 (e.g. in S3, S5, S6, etc.); Might the exact number of events be presented & analyzed instead?

10. For Supplementary Figure S6, unconventional OR (i.e. 416E3, 174E3) are provided, which might indicate a technical issue (e.g. division by zero) in computation of the values. This might be addressed.

11. For Supplementary Figures S2 to S12, while details for Markers 1, 2 and 3 are provided for each figure, it appears that only "1,2" is ever labelled for the PS-matched row for each comparator. If so, the authors might consider simply describing the propensity score criteria for PS-matched without requiring a Marker column.

12. There may be some grammatical/spelling issues, e.g. "the reproductive age"; "data collected form registers", "statistical significant findings" in the Abstract, "this tendency were reduced" (Page 10), etc. These might be addressed.

Reviewer #2: This is a study from Sweden and Denmark aiming to explore whether use of duloxetine in pregnancy could increase the risk of major and minor congenital malformation, and the risk of stillbirth. The authors linked data from nationwide registers from Sweden and Denmark covering all registered births from 2004-2016. Generally, this is an important research question although several previous studies have shown that use of duolexitine in pregnancy did not increase the risk of malformations and stillbirths, and thus the novelty of the study in the field is relatively limited. Also, some methodological aspects of the paper should be better addressed by the authors for a better interpretation of the findings. Here below are more specific comments on the various sections of the paper.

ABSTRACT:

in the "Exposure" paragraph you wrote "4) women discontinuing duloxetine treatment before pregnancy." Maybe you wanted to write during pregnancy?

INTRODUCTION:

You should review the English especially in the second paragraph. 

METHODS:

In the "Cohorts" paragraph you did not specify inclusion/esclusion criteria on age at delivery and gestational age. I recommend to you to consider such inclusion/exclusion criterias: ages at delivery between 12-55 years and gestational age between 22-42/46 gestational weeks. This latter, especially, is very important following the stillbirth outcome that you chose to evaluate. 

"Exposure, comparison groups and outcome" paragraph: 

(i) In the second paragraph I reccomad to specify the fourth comparison group as follow: "4) duloxetine discontinues: at least one redeemed prescription of duloxetine between 365 day priorLMP to LMP, BUT NOT DURING PREGNANCY, if it is the case. 

(ii) you stated that you excluded women with duloxetine exposure from 90 days prior to LMP, but no exposure from LMP to 90 days after LMP, why you made this choice if in the sensitivity analysis you evaluated the overlap days of supply? 

(ii) in the last sentence you define the stilbirth as a no signs of life at birth after week 22 of pregnancy, once again it is very important that all women included in the cohort have at least 22 weeks of gestation. 

"Statistical methods":

(i) in the third paragraph you said that "when fitting each model, covariates were removed, if the model could not be estimated.". Since you consider rare outcomes I recommend you to did not performe multivariate analysis but adjusted for the propensity score. I sow that you delete a huge number of women after the PS matching, why did you not performe the propensity score stratification? I would delete the multivariate analysis. 

(ii) you said that you assess the comorbidity in the 5 years prior to LMP, are you sure that all women have this follow-up prior to LMP? In case yes, you should add as inclusion/exclusion criteria. 

(iii) when you present the four sensitivity analysis you performed, I would say that you performed them to evaluate the effect of potential misclassification of exposure. 

(iv) in the third sensitivity analysis you stated that you restricted the cohort to the first pregnancy within the study, why did you not use the generalized estimating equation to account for potential correlation within women with multiple pregnancies during the considered period in the unadjusted analyses? You should assess in the undjusted analysis if the estimate with and without the generalized estimating equation to account for potential correlation within women with multiple pregnancies during the considered period are different, In case not you can delete the third sensitivity analysis. 

(v) you performed the last sensitivity analysis in complete case sample with BMI information, can you consider the missing for that variable as missing at random? 

RESULTS:

You shoal review the English and make all the paragraph more descriptive. 

TABLE and FIGURE:

(i) I would collapse the figure 2 and figure 3 in 1 Figure with Panel A and Panel B. You would have only the unadjusted and PS-matched (or I recommend to you to try the PS stratification) estimate for each comparison. 

(ii) I suggest to move the Table 2 in the supplementary materials

(iii) Table S6 I would take only the Post STD diff. 

Reviewer #3: This is a well-written revised manuscript that describes a study examining whether duloxetine exposure in pregnancy is associated with malformations or stillbirths. The study is well designed and controls for confounding by indication in several different ways which extremely important for this type of study. I have no suggestions for changes. Well done!!!

[LINK]

---

## [Decision Letter · Decision Letter 2]

24 Aug 2021

Dear Dr. Ankarfeldt,

Thank you very much for submitting your manuscript "Exposure to duloxetine during pregnancy and risk of congenital malformations and stillbirth: A nation-wide cohort study in Denmark and Sweden" (PMEDICINE-D-20-04640R2) for consideration at PLOS Medicine. 

Your paper was evaluated by a senior editor and discussed among all the editors here. It was also discussed with an academic editor with relevant expertise, and sent to two of the original reviewers, including a statistical reviewer. The reviews are appended at the bottom of this email and any accompanying reviewer attachments can be seen via the link below:

[LINK]

Although we will not be able to accept the manuscript for publication in the journal in its current form, we would like to consider a revised version that addresses the reviewers' and editors' comments. Obviously we cannot make any decision about publication until we have seen the revised manuscript and your response.

In revising the manuscript for further consideration, your revisions should address the specific points made by the reviewers and editors. Please also check the guidelines for revised papers at http://journals.plos.org/plosmedicine/s/revising-your-manuscript for any that apply to your paper. In your rebuttal letter you should indicate your response to the reviewers' and editors' comments, the changes you have made in the manuscript, and include either an excerpt of the revised text or the location (eg: page and line number) where each change can be found. Please submit a clean version of the paper as the main article file; a version with changes marked should be uploaded as a marked up manuscript.

We expect to receive your revised manuscript by Sep 14 2021 11:59PM. Please email us (plosmedicine@plos.org) if you have any questions or concerns.

We look forward to receiving your revised manuscript. 

Sincerely,

Caitlin Moyer, PhD

Associate Editor

PLOS Medicine

plosmedicine.org

1. Reviewer 2 Comment: Please do address the remaining comments of Reviewer 2. The editors feel that it is acceptable to present both multivariable and propensity-score matched analyses. Please do address the Reviewer’s request regarding restricting the analysis to those with at least 1 year of prior registry data. If this is not feasible we request that you include a thorough discussion of the limitation related to identifying a confounder in those with 5 years of back follow up in registry vs those with fewer years available in registry.

2. Data availability statement: For data that are freely or publicly available, please note this and provide more specific/direct links to access the Statistics Denmark and Statistics Sweden, and Swedish National Board of Health and Welfare data used in this study. For more information, please see the policy at

http://journals.plos.org/plosmedicine/s/data-availability

and FAQs at

http://journals.plos.org/plosmedicine/s/data-availability#loc-faqs-for-data-policy

3. Throughout text: Please include line numbers with the revised version.

4. Throughout text: Please use square brackets for in-text references, placed before the sentence punctuation. For multiple references within brackets, please do not include spaces.

5. Abstract: Please structure your abstract using the PLOS Medicine headings (Background, Methods and Findings, Conclusions).

6. Abstract: Please fully define all abbreviations at first use in the text (SNRI, SSRI, etc).

7. Abstract: Methods and Findings: In this section, please clearly describe the study design, population and setting, number of participants, years during which the study took place, length of follow up, and main outcome measures. Please quantify the main results (with 95% CIs and p values). Please note the important dependent variables that are adjusted for in the analyses. In the last sentence of the Abstract Methods and Findings section, please describe the main limitation(s) of the study's methodology.

8. Abstract: Conclusion: We suggest revising to “...no increased risk of major or minor congenital malformations or stillbirth was associated with exposure to duloxetine during pregnancy.”

9. Author summary: At this stage, we ask that you include a short, non-technical Author Summary of your research to make findings accessible to a wide audience that includes both scientists and non-scientists. The Author Summary should immediately follow the Abstract in your revised manuscript. This text is subject to editorial change and should be distinct from the scientific abstract. Please see our author guidelines for more information: https://journals.plos.org/plosmedicine/s/revising-your-manuscript#loc-author-summary

10. Introduction: The final paragraph of the section “The present study is based on a safety study…” seems more appropriate for the Methods section.

11. Methods: Please provide more description of the Prescription, Patient, and Medical Birth registers used in the study.

12. Methods: Where the four non-exposed groups are described, it might be helpful to mention that venlafaxine is another SNRI.

13. Methods: From Figure 3 and Table S8 it seems that adjusted analyses for the stillbirth outcome were not done. Please clarify this in the text, similar to: “For analyses of malformation subtypes with less than outcome events in the exposed group, only unadjusted and PS-matched analyses were performed.”

14. Methods: Did your study have a prospective protocol or analysis plan? Please state this (either way) early in the Methods section.

15. Methods: Please ensure that the study is reported according to the STROBE guideline, and include the completed STROBE checklist as Supporting Information. Please add the following statement, or similar, to the Methods: "This study is reported as per the Strengthening the Reporting of Observational Studies in Epidemiology (STROBE) guideline (S1 Checklist)."

16. Results: Please move the baseline table for stillbirth analyses to the main text.

17. Results: For all analyses described, please present the complete main results together with 95% CIs and p values. For example, please present the results for this analysis, with 95% CIs and p values. “For minor malformations, the unadjusted analysis of duloxetine exposed compared with duloxetine nonexposed showed an increased risk. However, in the adjusted and PS-matched analyses the risk was lower and showed no statistically significant increase.”

18. Results: Rather than describing “Similar patterns were observed in the sensitivity analyses.” for the stillbirth and minor malformations analyses, please describe what was observed.

19. Results: “It is of note that a statistically significant increased risk of “other anomalies and syndromes” was found when duloxetine exposed women were compared with SSRI exposed. The PS-matched analyses yielded OR 2.43 (95% CI 1.10-5.38).” Please make it more clear that the PS-matched result presented here is for the SSRI-exposed comparison.

20. Discussion: Please present and organize the Discussion as follows: a short, clear summary of the article's findings; what the study adds to existing research and where and why the results may differ from previous research; strengths and limitations of the study; implications and next steps for research, clinical practice, and/or public policy; one-paragraph conclusion.

21. Discussion: Please clarify whether this refers to increased risk of malformation or stillbirth: “Like the present study, a small Swedish study analyzing the association between SNRI/NRIs (not including duloxetine) and stillbirth found no increased risk.”

22. Discussion: We suggest replacing “compliance” with “adherence” when referring to recommended folic acid supplementation.

23. Discussion: Generalizability: In this paragraph, we suggest a broader discussion on clinical or policy implications and further directions based on the findings reported here.

24. Disclosure of conflicts of interest: These can be removed from the main text of the manuscript, and please make sure all information is completely and accurately entered in to the manuscript submission metadata.

25. References: Please use the "Vancouver" style for reference formatting, and see our website for other reference guidelines https://journals.plos.org/plosmedicine/s/submission-guidelines#loc-references

26. Reference List: Reference 49: Papers cannot be listed in the reference list until they have been accepted for publication or are publicly available on a preprint archive. The information may be cited in the text as a personal communication with the author if the author provides written permission to be named. Alternatively please provide a different appropriate reference.

27. Reference 51: Please provide the complete citation information.

28. Figure 1: Please define all abbreviations in the legend (SSRI, LMP).

29. Figure 2 and Figure 3: Please define all abbreviations in the legend (SSRI).

30. Table S5: Please move the table to the main section of the paper.

31. All tables: Please make sure to define all abbreviations used in the tables in the legends.

Comments from the reviewers:

Reviewer #1: We thank the authors for addressing the previous issues raised, and have no further concerns.

Reviewer #2: Dear Editor and Authors,

my comment is regarding only the statistical methodology. I think that it has no sense to perform both the multivariate and ps matching adjustment, especially in this case where the prevalence of the outcomes are very rare. 

The PS's methodology is, in fact, a methodology used to (i) avoid the overfitting problem (you should record at least 8 events for each confounder included in the model), and (ii) compare exposed and unexposed comparable.

Moreover, you should consider that if you do not require that all women have at least 5 years back-follow up available, the probability to find a confounder in women with 5 years vs those with fewer years is different. Since you considered chronic conditions and you assessed the concomitant medications only in the year before LMP I will recommend you to ask as inclusion criteria that all women have at least one year prior to the LMP and to evaluate the confounders in this time window.

[LINK]

---

## [Editor Report · Decision Letter 3]

4 Oct 2021

Dear Dr. Ankarfeldt,

Thank you very much for re-submitting your manuscript "Exposure to duloxetine during pregnancy and risk of congenital malformations and stillbirth

A nation-wide cohort study in Denmark and Sweden" (PMEDICINE-D-20-04640R3) for review by PLOS Medicine.

I have discussed the paper with my colleagues and the academic editor. I am pleased to say that provided the remaining editorial and production issues are dealt with we are planning to accept the paper for publication in the journal.

[LINK]

In revising the manuscript for further consideration here, please ensure you address the specific points made by the editors. In your rebuttal letter you should indicate your response to the editors' comments and the changes you have made in the manuscript. Please submit a clean version of the paper as the main article file. A version with changes marked must also be uploaded as a marked up manuscript file.

We look forward to receiving the revised manuscript by Oct 11 2021 11:59PM.   

Sincerely,

Caitlin Moyer, Ph.D.

Associate Editor 

PLOS Medicine

plosmedicine.org

Requests from Editors:

1. Title: Please format the title as: “Exposure to duloxetine during pregnancy and risk of congenital malformations and stillbirth: A nation-wide cohort study in Denmark and Sweden”

2. Data availability statement: Please note that the link provided for the Swedish National Board of Health and Welfare data did not work, and there may be a typo in the link. If accurate please replace with: https://www.socialstyrelsen.se/en/statistics-and-data/statistics/statistical-databases/

We suggest revising slightly to: “Data from the Danish and Swedish registers are third party data, meaning that we as researchers do not hold the data, but have obtained data after application at relevant parties. The Danish data can be applied for at Statistics Denmark (https://www.dst.dk/en/TilSalg/Forskningsservice). The Swedish data

can be applied for at Statistic Sweden (https://www.scb.se/en/About-us/contact-us/) and Swedish National Board of Health and Welfare data (https://www.socialstyrelsen.se/en/statistics-and-data/statistics/statistical-databases/).

3. Abstract: Methods and Findings: Line 45: We suggest revising to “A population-based observational study was conducted based on data from registers in Sweden and Denmark.” or similar, to make this a complete sentence.

4. Abstract: Methods and Findings: Line 59-62: Please revise this sentence to: “Duloxetine exposed vs duloxetine non-exposed propensity score matched analyses showed odds ratios (OR) of 0.98 (95% confidence interval [CI] 0.74-1.30) for major malformations, 1.09 (95% CI 0.82-1.45) for minor malformation, and 1.18 (95% CI 0.43-3.19) for stillbirths.” or similar to clarify. Please also include p values in addition to reporting the OR and confidence intervals.

5. Abstract: Methods and Findings: Line 63-65: Please clarify this sentence, such as: “The main limitations for the study were that the indication for duloxetine and a direct measurement of depression severity were not available to include as covariates.” if this is what was meant.

6. Author summary: Why was this study done? We suggest combining the first two points, such as: “Many women of reproductive age take drugs used to treat depression, including duloxetine, a selective serotonin-norepinephrine reuptake inhibitor (SNRI) approved in 2004 for the treatment of depression, and the use of which has been increasing.” or similar.

7. Author summary: What did the researchers do and find? We suggest combining the second and third bullet points: “From more than 2 million births identified, information on drug exposure, comorbidities, education and income, and congenital malformations and stillbirths was gathered.” or similar.

8. Author summary: What did the researchers do and find? We suggest revising the third bullet point to: “The analyses taking into account factors beyond duloxetine exposure did not reveal associations between exposure to duloxetine during pregnancy and risk for malformations or stillbirth.” or similar.

9. Methods: Study protocol: Thank you for including the link to the protocol on the ENCePP website. We request that you include a copy of the protocol as a supporting information file. Please remove confidential information or confidentiality statements and any trademark/copyright symbols. Also, please clarify that the online protocol also seems to suggest that preterm birth and small for gestational age would be included in the analyses, and describe when and the rationale for the change in protocol.

10. Methods: Line 227: We suggest renaming the STROBE checklist file to "S1 Checklist" or similar.

11. Methods: Line 228: We suggest revising to “The prospectively developed protocol was followed…” if this is accurate. At line 232 please revise to “...comedication should have been identified one year prior to LMP, but was changed to comedication during the relevant exposure time window…” if accurate.

12. Results: Line 245-247,Lines 254-257, Lines 262-264, Lines 266-267, and Lines 269-272: Please also present the p values for the PS-matched results in the text.

13. Discussion: Line 291-294: It may be helpful to briefly mention some of the “other anomalies and syndromes” that would fall into this category.

14. Discussion: Line 295-296: We suggest providing some additional context for this previously published study investigating stillbirth risk, and expanding to discuss your findings and interpretations in light of this.

15. Discussion: Line 340: Please revise to “...the number of women exposed to duloxetine during pregnancy was limited…” if this is accurate.

16. Line 360-365: Please remove the data availability statement from the main text, and please ensure that the data availability statement of the manuscript submission is complete and accurate.

17. References: Please use the "Vancouver" style for reference formatting, and see our website for other reference guidelines https://journals.plos.org/plosmedicine/s/submission-guidelines#loc-references

For example, in reference 3 the journal title should be abbreviated as “Obstet Gynecol” and it seems as if “[Article]” could be removed from reference 54. In addition, the link provided for reference 46 does not seem to work. Please check each individual reference for accuracy, journal title abbreviations and all formatting throughout.

18. Figure 2 and Figure 3, Table 1 and Table 2: In the legends, please correct “LPM” to “LMP” if this is accurate.

19. Figure 2 and 3 Legends: Please note that the legends on page 21 refer to covariate information presented in Table S6, while this information appears to be presented in Table S5.

20. Supporting information file: In the “Contents” it seems as if Table S8 listed as being on page 17 is a typo and should be Table S6. Also please note that there are two versions of Tables S6 and S7 included.

21. Supporting information Table S6 and S7. Please change the reference to the list of covariates in the legends to Table S5 rather than Table S6.

[LINK]

---

## [Editor Report · Decision Letter 4]

15 Oct 2021

Dear Dr Ankarfeldt, 

On behalf of my colleagues and the Academic Editor, Jenny E Myers, I am pleased to inform you that we have agreed to publish your manuscript "Exposure to duloxetine during pregnancy and risk of congenital malformations and stillbirth: A nation-wide cohort study in Denmark and Sweden" (PMEDICINE-D-20-04640R4) in PLOS Medicine.

Please also address the following editorial requests:

1. Abstract: Line 50: Please remove "sex" from the abstract as this does not appear to have been included as a confounder variable.

2. Discussion: Line 326: Please change "comply with" to "adhere to" in this sentence.

3. Supporting Information File: Please remove the box with "Confidential Information" from page 1 of the Protocol document included with the supporting information.

PRESS

Sincerely, 

Caitlin Moyer, Ph.D. 

Associate Editor 

PLOS Medicine